# Potassium Intake—(Un)Expected Non-Predictor of Higher Serum Potassium Levels in Hemodialysis DASH Diet Consumers

**DOI:** 10.3390/nu14102071

**Published:** 2022-05-15

**Authors:** Cristina Garagarza, Ana Valente, Cristina Caetano, Inês Ramos, Joana Sebastião, Mariana Pinto, Telma Oliveira, Aníbal Ferreira, Catarina Sousa Guerreiro

**Affiliations:** 1Nutrition Department, Nephrocare, 1250-191 Lisbon, Portugal; ana.valente@fmc-ag.com (A.V.); cristina.caetano@fmc-ag.com (C.C.); ines.ramos@fmc-ag.com (I.R.); joana.sebastiao@fmc-ag.com (J.S.); mariana.pinto@fmc-ag.com (M.P.); telma.oliveira@fmc-ag.com (T.O.); 2Nutrition Laboratory, Faculty of Medicine, Lisbon University, 1649-004 Lisbon, Portugal; cfguerreiro@medicina.ulisboa.pt; 3Nephrology Department, Dialysis Unit Vila Franca de Xira, 2600-076 Vila Franca de Xira, Portugal; anibal.ferreira@netcabo.pt; 4Faculty of Medical Sciences, Nova Medical School, 1169-056 Lisbon, Portugal; 5Institute of Environmental Health, Faculty of Medicine, Lisbon University, 1649-004 Lisbon, Portugal

**Keywords:** dietary intake, DASH diet, hemodialysis

## Abstract

As high serum potassium levels can lead to adverse outcomes in hemodialysis (HD) patients, dietary potassium is frequently restricted in these patients. However, recent studies have questioned whether dietary potassium really affects serum potassium levels. The dietary approaches to stop hypertension (DASH) diet is considered a healthy dietary pattern that has been related to lower risk of developing end-stage kidney disease. The aim of this study was to analyze the association between a dietary pattern with high content of potassium-rich foods and serum potassium levels in HD patients. This was an observational, cross-sectional, multicenter study with 582 HD patients from 37 dialysis centers. Clinical and biochemical data were registered. Dietary intake was obtained using the Food Frequency Questionnaire. Adherence to the DASH dietary pattern was obtained from Fung’s DASH index. All statistical tests were performed using SPSS 26.0 software. A *p*-value lower than 0.05 was considered statistically significant. Patients’ mean age was 67.8 ± 17.7 years and median HD vintage was 65 (43–104) months. Mean serum potassium was 5.3 ± 0.67 mEq/L, dietary potassium intake was 2465 ± 1005 mg/day and mean Fung´s Dash Index was 23.9 ± 3.9. Compared to the lower adherence to the DASH dietary pattern, patients with a higher adherence to the DASH dietary pattern were older (*p* < 0.001); presented lower serum potassium (*p* = 0.021), serum sodium (*p* = 0.028), total fat intake (*p* = 0.001) and sodium intake (*p* < 0.001); and had higher carbohydrate intake (*p* < 0.001), fiber intake (*p* < 0.001), potassium intake (*p* < 0.001), phosphorus intake (*p* < 0.001) and body mass index (*p* = 0.002). A higher adherence to this dietary pattern was a predictor of lower serum potassium levels (*p* = 0.004), even in the adjusted model (*p* = 0.016). Following the DASH dietary pattern, which is rich in potassium, is not associated with increased serum potassium levels in HD patients. Furthermore, a higher adherence to the DASH dietary pattern predicts lower serum potassium levels. Therefore, generalized dietary potassium restrictions may not be adequate, at least for those with a DASH diet plan.

## 1. Introduction

In hemodialysis (HD) patients, serum potassium is frequently monitored and managed to maintain values between 3.5 and 5.5 mmol/L [1,2]. Due to impaired renal excretion, these patients are more prone to developing hyperkalemia and, therefore, suffer its consequences. Mild hyperkalemia (5.5–5.9 mmol/L) may be associated with symptoms such as nausea, fatigue, or muscle weakness, but severe hyperkalemia (≥6.5 mmol/L) can cause alterations in cardiac physiology, leading to chest pain, cardiac arrhythmias, shortness of breath, and fatal cardiac arrest [3,4,5].

Apart from the impaired renal potassium excretion related to kidney failure, hyperkalemia in HD patients may result from other clinical conditions such as diabetes mellitus, metabolic acidosis, constipation, medications (potassium-sparing diuretics, beta-blocking agents, angiotensin II receptor blockers, angiotensin-converting enzyme inhibitors and non-steroidal anti-inflammatory drugs) [6,7]. 

As high serum potassium levels can cause adverse outcomes in HD patients, different guidelines suggest that patients should restrict their dietary potassium intake but the evidence supporting this restriction independent of its food sources in order to improve morbidity, mortality and quality of life in the HD population is limited [8,9].

However, patients in maintenance HD are frequently instructed to reduce their dietary potassium intake to prevent high serum potassium levels or in response to altered laboratory results [10]. This recommendation focuses mainly on limiting the consumption of fruits, vegetables, legumes, whole grains, nuts and processed foods. Recently, some authors have questioned this approach and whether dietary potassium and, specially, its food source affects serum potassium levels [11]. Rather than concentrating only on the potassium amount in foods, the type of food and its content in other nutrients should be considered when assessing the impact on serum potassium.

The dietary approaches to stop hypertension (DASH) diet is considered a healthy dietary pattern that has been related to lower risk of developing end-stage renal disease [12]. It emphasizes the consumption of potassium-rich foods, especially from plant sources, such as fruits, vegetables, whole grains, nuts and seeds. Moreover, it advocates reduced intakes of sodium, sugar-sweetened beverages, and red and processed meat. In our study the aim was to analyze the relationship between a dietary pattern rich in high-potassium foods (DASH) and serum potassium in HD patients.

## 2. Materials and Methods

### 2.1. Study Design and Setting

This was an observational, cross-sectional, multicenter study with 582 HD patients from 37 dialysis centers.

### 2.2. Sample Size

Among the 4600 patients undergoing HD in 37 dialysis centers, 600 patients were selected; patients fulfilling the inclusion criteria were randomly selected equally from each dialysis center, and 18 patients refused to participate in the study (3%). Therefore, we collected data from 582 patients.

### 2.3. Inclusion and Exclusion Criteria

Patients were eligible for this study if they were aged ≥ 18 years, had undergone 4 h in-center HD sessions 3 times a week for ≥15 months (with an online hemodiafiltration technique), had been accepted to participate, and had signed an informed consent. 

All patients were dialyzed with high-flux membranes (Helixone^®^, Fresenius^®^ Medical Care, Bad Homburg, Germany) and ultrapure water in accordance with the criteria of ISO regulation 13,959:2009—Water for hemodialysis and related therapies. Patients were ineligible if they met any of the following criteria: low comprehension of the country language, severe neurological or mental disorder, active neoplastic disease, major amputation (lower/upper extremities), enteral or parenteral feeding, severe alcohol or drug addiction, hepatitis C with viral replication, liver disease, and immunosuppressive or corticoid medication. All the patients in our study had been given dietary recommendations in line with current dietary guidelines for dialysis patients at the initiation of the HD treatment.

### 2.4. Data Analysis

Demographic, anthropometric, biochemical and dialysis treatment data were obtained from the dialysis units database in the same month as the face-to-face interviews. We collected blood for the biochemical analysis before the midweek HD session. All the laboratory measures were tested using identical methods in different laboratories. 

### 2.5. Food Frequency Questionnaire (FFQ)

We assessed dietary intake through a semi-quantitative FFQ conducted by a dietitian in a face-to face interview during the HD treatment. It had been developed and validated for the Portuguese population [13,14] It had 95 food items, 9 categories of frequencies (from “never or less than once a month” to “six or more times a day”), and a section with predetermined average portions. The frequency of intake and the mean portions of each food item were registered and illustrated through a book with 131 colored photos, serving as a visual auxiliary for the patients. The respondent was asked to describe her or his diet over the last 1-year period. To estimate dietary intake, the frequency reported for each item was multiplied by the respective portion (in grams) and by a factor for seasonal variation of food items that are eaten in specific times during the year. This questionnaire gives information regarding the average daily amount of macro- and micronutrients consumed. The conversion of food item into nutrients was carried out with the Food Processor Plus software (ESHA Research, Salem, Oregon) containing the nutritional data from the United States Department of Agriculture and adapted to typical Portuguese foods. The nutrient content of Portuguese foods was added to the original database using the Portuguese food composition Table 1 [15]. For the data analysis, food items with a mean intake ≤5 g/day were excluded. 

Food groups were created according to the components of the DASH index. The adherence to this dietary pattern was obtained from Fung’s DASH index (8–40 points) [16]. The DASH diet index developed by Fung et al. [17] (9) consists of eight items (seven food groups and one nutrient) based on foods and nutrients more or less relevant in the DASH diet according to the eating recommendations developed by the National Heart, Lung and Blood Institute [18]. The index scores sex-specific quintile rankings of eight food components (servings per day) for recommended components such as intakes of fruit (includes fruit juice); vegetables (excludes potatoes); low-fat dairy products; whole grains; and nuts, seeds, and legumes. Scores from 1 (lowest quintile) to 5 (highest quintile) are attributed to patients. On the contrary, individuals receive scores from 1 (highest quintile) to 5 (lowest quintile) for components for which lower intakes are desirable, such as sodium, sugar-sweetened beverages, and red and processed meat. Items’ scores are summed to a total DASH score that ranges from 8 to 40 points (Table 1).

### 2.6. Statistical Analysis

Categorical variables were presented as frequency (percentages) and continuous variables were presented as mean ± standard deviation (SD) or as median and interquartile ranges (IQR). Data distribution was tested with Kolmogorov–Smirnov test. Pearson’s correlation was used to analyze the correlation between serum potassium and dietary potassium intake, serum potassium and food intake, and dietary potassium and food intake.

For the statistical analysis, Fung’s DASH index was categorized into terciles. Therefore, the sample was divided into 3 groups depending on their adherence to this dietary pattern.

Mean differences were evaluated using one-way ANOVA for variables normally distributed and the Kruskal–Wallis test for variables not normally distributed. The categorical variables were analyzed using the Pearson’s chi-squared test. 

The effect of the adherence to the DASH dietary pattern (as an independent variable) on serum potassium levels was tested with a linear regression analysis. The multiple linear regression model was adjusted for age, gender, presence of diabetes mellitus, energy intake, dietary potassium intake, residual diuresis, dialysis adequacy (Kt/V), dialysis vintage and intake of potassium binders.

Statistical analysis was run with the SPSS software (version 26.0; IBM SPSS, Inc., Chicago, IL, USA), and a *p*-value <  0.05 was considered statistically significant.

## 3. Results

Patients’ mean age was 67.8 ± 17.7 years and median HD vintage was 65 (43–104) months. From the whole sample, 41.4% (*n* = 241) were female and 31.6 % (*n* = 184) had diabetes mellitus. Mean serum potassium was 5.3 ± 0.67 mEq/L, and mean dietary potassium intake was 2465 ± 1005 mg/day. 

We did not observe statistically significant correlation between serum potassium and dietary potassium intake (r = 0.080; *p* = 0.060) (Figure 1). The same correlation analysis was run after separating patients with lower potassium intakes (≤3000 mg/day) and higher potassium intakes (>3000 mg/day), but still no statistically significant correlation between serum potassium and dietary potassium intake was observed in any group: lower potassium intake group (*n* = 418; r = 0.056; *p* = 0.253); higher potassium intake group (*n* = 126; r = −0.031; *p* = 0.731). Furthermore, no differences were observed in serum potassium means between these two groups: serum potassium in the lower potassium intake group = 5.2 ± 0.69 mEq/L and serum potassium in the higher potassium intake group = 5.4 ± 0.60 mEq/L (*p* = 0.061). 

However, foods that showed a positive correlation with serum potassium levels were milk (r = 0.121; *p* = 0.005); eggs (r = 0.090; *p* = 0.037); beef, pork and chicken liver (r = 0.009; *p* = 0.037); fatty fish (r = 0.122; *p* = 0.004); squid and octopus (r = 0.086; *p* = 0.045); banana (r = 0.090; *p* = 0.036); canned fruit (r = 0.099; *p* = 0.021); wine (r = 0.091; *p* = 0.034); and coffee (r = 0.086; *p* = 0.046).

On the other hand, foods with higher positive correlation (≥0.300) with dietary potassium intake were boiled potato (r = 0.424; *p* < 0.001), cow and pork meat (r = 0.407; *p* < 0.001), white cabbage (r = 0.402; *p* < 0.001), apple and pear (r = 0.397; *p* < 0.001), cherry (r = 0.374; *p* < 0.001), yogurt (r = 0.365; *p* < 0.001), orange (r = 0.340; *p* < 0.001), beans (r = 0.335; *p* < 0.001), peach (r = 0.335; *p* < 0.001), tomato (r = 0.331; *p* < 0.001) and milk (r = 0.323; *p* < 0.001).

The mean Fung’s Dash Index was 23.9 ± 3.9. Of the whole sample, 36.4% of the patients presented low adherence to the DASH dietary pattern (0–22 points), 39.0% presented a moderate adherence (23–26 points), and 24.6% a high adherence (≥27 points). 

Compared to the lower adherence group, patients with a higher adherence to the DASH dietary pattern were older (*p* < 0.005); presented lower serum potassium (*p* = 0.021), serum sodium (*p* = 0.001), albumin (*p* = 0.030), energy intake (*p* = 0.006), % of fat intake (*p* = 0.001) and sodium intake (*p* < 0.005); had a higher prevalence of diabetes (<0.001) and higher body mass index (BMI) (*p* = 0.002); and had higher intakes of carbohydrates (%DEI) (<0.001), fiber (<0.001), potassium (<0.001), phosphorus (<0.001) and calcium (*p* < 0.001) (Table 2). 

Regarding dialysis-related parameters (HD vintage, potassium binders, Kt/V, interdialytic weight gain (%IDWG) and diuresis), no differences were observed among groups (Table 2).

A higher adherence to this dietary pattern was a predictor of lower serum potassium levels (*p* = 0.004), even in the adjusted model (*p* = 0.016) (Table 3).

## 4. Discussion

Except from the recent Nutrition KDOQI Guidelines [19], which suggest an individualization of the dietary potassium intake recommendation, older clinical practice guidelines in the field of renal nutrition suggest a threshold of 3000 mg for the maximum daily potassium intake to avoid hyperkalemia and, therefore, to improve outcomes [1,20,21].

In this study, we found no association between higher dietary potassium intake and higher serum potassium levels. Furthermore, patients with a higher adherence to the DASH dietary pattern, which is characterized as being rich in potassium and low in sodium, presented a mean potassium intake of 2765 mg/day but lower serum potassium and sodium levels. Moreover, following a DASH diet was a predictor of lower serum potassium in the adjusted model. 

The association between dietary potassium intake and its serum levels is controversial. Potassium salts, used as additives, have been shown to result in post-prandial serum potassium increase in patients with chronic kidney disease, but dietary potassium intake appears to be weakly associated with pre-dialysis serum potassium in HD patients. Noori et al. observed a positive, marginal correlation (r = 0.14, *p* < 0.05) of dietary potassium with pre-dialysis serum levels and hypothesizes that this relationship is only strong in cases of very low or very high intakes [22]. However, we did not observe any correlation between dietary potassium and serum potassium after separating patients into two groups of potassium intake (≤3000 mg/day and >3000 mg/day).

Other authors have found similar results to those observed in our study, as no association was registered between serum and dietary potassium [23,24]. However, the inexistence of a positive correlation between these two parameters does not mean that high- potassium foods do not contribute to hyperkalemia in these patients, but different sources of dietary potassium (animal, plant, or potassium-based food additives) may differently contribute to hyperkalemia due to potassium bioavailability. González-Ortiz et al. found that HD patients consuming a healthy plant-based diet did not present higher serum potassium or hyperkaliemia compared with patients not following a healthy plant-based diet [2]. However, we should take into account that demineralization of foods, especially vegetables, by different cooking methods, can reduce their potassium content [2].

In our study, except for banana, canned fruit, wine and coffee, the foods which were positively correlated with serum potassium levels belonged mainly to animal sources (milk, beef, pork and chicken liver, fatty fish, squid and octopus).

Without an adequate dietary advice, restricting all high-potassium foods can result in a poorly diversified diet and reduced consumption of foods containing other important nutrients such as vitamins, minerals and fiber. Furthermore, nutrient interactions must be considered when managing dietary potassium intake and serum potassium levels as there are other nutrients in food (e.g., fiber) that can affect potassium distribution and excretion [24]. 

In this study, patients with the highest adherence to the DASH dietary pattern showed the highest intake of carbohydrates (55.5 % of the DEI) and the lowest total fat intake (27.9% of the DEI). We believe that these findings are mainly due to the macronutrients that characterize part of the food groups of the DASH diet (fruits, vegetables and grains), which are rich in carbohydrates and present low amounts of fat. 

On the one hand, the DASH dietary pattern emphasizes the intake of plant-based foods for which potassium bioavailability is known to be lower [25,26]. On the other hand, this increased carbohydrates intake contributes to a high intake of fiber, which in turn, leads to potassium excretion [24]. Carbohydrate-rich foods that are also rich in potassium may have a lower effect on serum potassium than foods that are low in carbohydrates and high in potassium. Food options with high carbohydrate content lead to insulin release, which can attenuate the initial increase in serum potassium [11].

Therefore, patients with the highest adherence to the DASH dietary pattern may benefit from this diet for a better control of serum potassium levels. The levels of intake of fiber were significantly different among the three groups of patients, with the more adherent patients presenting a median daily intake of 26.3 g (19.4–33.1) versus 17.2 g (13.3–22.8) in the lowest tercile and being the only group of patients with a fiber intake above the recommendations of 25–35 g/day [27]. 

Hayes et al. demonstrated that potassium elimination in stool was three times higher in the HD population in comparison to normal controls, reaching almost 80% of dietary potassium in some cases. Furthermore, fecal potassium content was directly proportional to dietary potassium intake and stool weight [28]. As the gastrointestinal tract represents another pathway for potassium elimination [29,30], it is extremely important to follow a dietary pattern including an adequate consumption of fiber, such as the DASH dietary pattern, due to its role in stimulating bowel movements. Moreover, constipation, instead of potassium dietary load, has been considered the main determinant of high potassium levels in individuals with end-stage kidney disease [24].

Along with fiber, the DASH dietary pattern is characterized by additional factors that may help avoid a rise in serum potassium, such as the significant consumption of alkali foods (fruits and vegetables) which may facilitate the intracellular movement of potassium, mostly in the presence of metabolic acidosis. Prolonged infusions of sodium bicarbonate in HD patients with hyperkalemia decreased serum potassium [31]. The biological processes associating acidosis and hyperkalemia appear to involve a complex interaction of numerous ion transporters, which contribute to maintaining the blood pH balance by indirectly stimulating an exchange of H^+^ for K^+^ between intracellular and extracellular compartments [31,32]. In this study, despite the higher levels of serum bicarbonate observed in the highest tercile of adherence to the DASH dietary pattern, no statistically significant difference was noted. We believe that the small variation in this parameter observed among groups may explain this finding.

Compared to the lower adherence group, patients with a higher adherence to the DASH dietary pattern were older and presented a higher prevalence of diabetes and a more elevated BMI (26.9 kg/m^2^ (23.8–31.1)). Regarding BMI, Smyth et al. found different results as patients in the highest quintile of the DASH dietary pattern were those with a lower BMI (27.3 ± 5.2 vs. 26.5 ± 4.9 kg/m^2^) [33]. We believe that our findings are related to the higher prevalence of DM also observed in this group as these patients usually present an increased BMI [34]. 

Another well-known factor that contributes to hyperkalemia is the presence of DM. Ramos et al. [23] observed that DM contributed to a rise in serum potassium levels (OR 4.22 (95% CI 1.31–13.6)). Reduced renal and intestinal elimination of potassium as well as impaired intracellular storage caused by insulin deficiency increase this risk [35,36]. Moreover, the typical diabetic diet is often rich in fruits, vegetables and whole grains, presenting a high potassium content. Despite the higher prevalence of patients with DM in the highest tercile of adherence to the DASH dietary pattern, this tercile presented lower levels of serum potassium levels. Furthermore, data from our multivariate analysis demonstrate that a high adherence to the DASH dietary pattern (inferring a higher consumption of potassium) predicted lower levels of this micronutrient, independently of the presence of DM.

In summary, we believe that there are important nutritional factors in the DASH diet pattern, such as the high fiber content and its alkali profile, contributing to our findings.

As strengths, we consider the sample size obtained from 37 dialysis clinics representative of several geographical zones in the country, which gives us a wide perspective. Another strong point is the fact that, to date, very few studies have been published in this field but interest in the role of serum potassium management through diet is re-emerging. As limitations, we point out the fact that we used retrospective questionnaires, where answers depend on the precision of patients’ recollections as well as the absence of data regarding other medication that can affect serum potassium levels.

## 5. Conclusions

Following the DASH dietary pattern, characterized by potassium-rich foods, is not associated with increased serum potassium levels in HD patients. Furthermore, a higher adherence to the DASH dietary pattern predicts lower serum potassium levels. Therefore, tight dietary potassium restrictions to manage hyperkalemia may not be adequate, at least for those with a DASH diet plan. A more flexible diet, especially regarding plant-based foods, may promote other health benefits and reduce patient’s dietary plan limitations.

Apart from restricting dietary intake of potassium from selected food sources, other conditions that influence serum potassium balance should be considered to control high serum potassium levels in HD patients. 

In summary, our study highlights the importance of dietary interventions for the management of serum potassium in HD patients and suggests a potential role of the DASH dietary pattern in controlling serum potassium levels and, therefore, limit hyperkalemia risk in these patients.

Finally, we believe that studies targeting other potential benefits of a DASH dietary approach in the HD field are necessary. 

## Figures and Tables

**Figure 1 nutrients-14-02071-f001:**
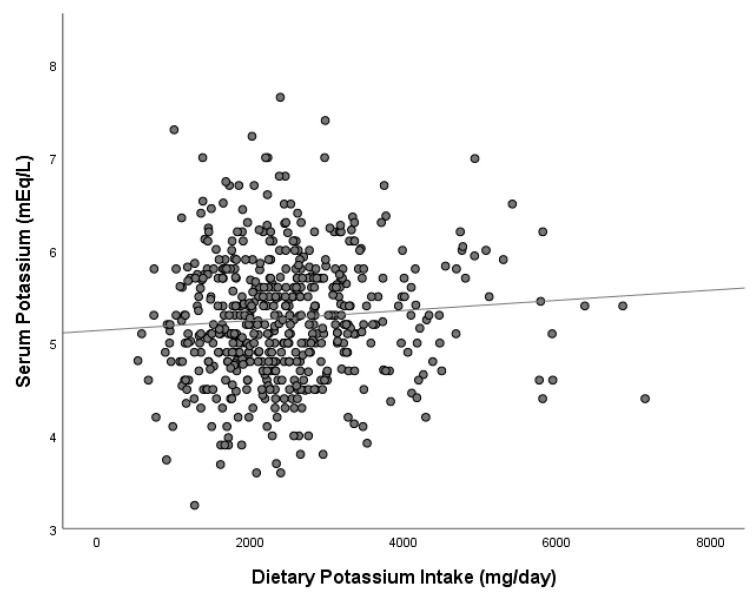
Correlation between serum potassium and dietary potassium intake (r = 0.08; *p* = 0.06).

**Table 1 nutrients-14-02071-t001:** Standards for scores on Fung’s DASH diet index.

Individual Components	Fung’s Dash Index (Sex Specific)	Score
Total Fruit	Fifth quintile	1—lowest quintile → to 5—highest quintile.
Vegetables (Excluding potatoes)	Fifth quintile	1—lowest quintile → to 5—highest quintile.
Whole grains	Fifth quintile	1—lowest quintile → to 5—highest quintile.
Low-fat dairy products	Fifth quintile	1—lowest quintile → to 5—highest quintile.
Nuts, seeds and legumes	Fifth quintile	1—lowest quintile → to 5—highest quintile.
Red and Processed meat	First quintile	1—highest quintile → to 5—lowest quintile.
Sugar-sweetened beverages	First quintile	1—highest quintile→ to 5—lowest quintile.
Sodium	First quintile	1—highest quintile → to 5—lowest quintile.
Total score (points)	8–40	

**Table 2 nutrients-14-02071-t002:** Mean differences between terciles of adherence to the DASH dietary pattern.

Parameter	Low Adherence (T1; *n* = 212)	Moderate Adherence (T2; *n* = 227)	High Adherence (T3; *n* = 143)	*p*
Age (years)	65 ± 16	68 ± 13	71 ± 10	**<0.001**
Gender—Women (%)	32.1	42.7	53.8	**<0.001**
Diabetes Mellitus (%)	24.5	28.6	46.9	**<0.001**
BMI (Kg/m^2^)	25.0 (22.1–28.2)	25.5 (22.5–28.8)	26.9 (23.8–31.1)	**0.002**
**Biochemical parameters**				
Potassium (mEq/L)	5.3 ± 0.7	5.3 ± 0.6	5.1 ± 0.7	**0.021**
Phosphorus (mg/dL)	4.3 ± 1.2	4.4 ± 1.2	4.2 ± 1.1	0.277
Sodium (mmol/L)	139.5 ± 2.9	139.2 ± 2.6	138.6 ± 2.9	**0.028**
Bicarbonate (mEq/L)	21.9 ± 2.0	21.8 ± 1.8	22.2 ± 2.0	0.784
Calcium (mg/dL)	8.9 ± 0.7	9.0 ± 0.8	8.9 ± 0.7	0.333
Hemoglobin (g/dL)	11.2 (10.6–12.0)	11.3 (10.6–11.8)	11.2 (10.7–11.8)	0.870
Albumin (g/dL)	4.1 (3.9–4.3)	4.1 (3.9–4.3)	4.0 (3.8–4.2)	**0.030**
C-reactive Protein (mg/L)	10.3 ± 12.2	9.3 ± 14.3	13.0 ± 17.3	0.559
**Dietary intake**				
Energy intake (Kcal/day)	1903 (1496–2362)	1668 (1347–2082)	1873 (1532–2411)	**0.001**
Protein (%E)	16.9 (15.4–18.7)	16.7 (15.3–19.1)	16.5 (15.2–17.8)	0.208
Total Fat (%E)	30.5 (26.4–33.9)	29.0 (25.1–32.4)	27.9 (24.9–31.7)	**0.001**
Carbohydrates (%E)	51.2 (46.0–56.3)	52.8 (47.7–56.8)	55.5 (51.2–59.0)	**<0.001**
Fiber (g/day)	17.2 (13.3–22.8)	18.1 (13.8–23.5)	26.3 (19.4–33.1)	**<0.001**
Phosphorus (mg/day)	1091 (842–1379)	980 (762–1282)	1169 (873–1533)	**<0.001**
Potassium (mg/day)	2249 (1774–2808)	2170 (1694–2662)	2765 (2115–3453)	**<0.001**
Sodium (mg/day)	2469 (1904–3157)	1978 (1526–2603)	2106 (1618–2717)	**<0.001**
Calcium (mg/day)	647 (442–915)	601 (443–828)	749 (545–1018)	**<0.001**
**Dialysis-related parameters**				
HD vintage (months)	65 (42–112)	69 (44–104)	63 (42–91)	0.340
Potassium binders (mg/week)	10.2 ± 37.7	11.6 ± 35.1	22.7 ± 141.2	0.280
Kt/V	1.68 (1.47–1.88)	1.72 (1.50–1.88)	1.72 (1.49–1.96)	0.688
IDWG (%)	3.2 (2.4–4.1)	3.1 (2.3–4-1)	3.0 (2.2–3.9)	0.231
Diuresis ≥200 mL/day—Yes (%)	55.7	58.0	67.6	0.068

**Table 3 nutrients-14-02071-t003:** Linear regression between serum potassium and adherence to the DASH index.

Dependent Variable	Independent Variable	*B*	95% CI_a_ *	*p*
**Serum potassium**	Adherence to DASH Index	−0.026	(−)0.047–(−0.005)	**0.016**

* Multivariate model adjusted for age, gender, presence of diabetes mellitus, energy intake, dietary potassium intake, residual diuresis, dialysis adequacy (Kt/V), dialysis vintage and intake of potassium binders.

## Data Availability

The data presented in this study are available on request from the corresponding author. The data are not publicly available due to privacy issues.

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
