# Peer review of "Potassium Intake—(Un)Expected Non-Predictor of Higher Serum Potassium Levels in Hemodialysis DASH Diet Consumers"

_nutrients, 2022, doi:10.3390/nu14102071_

Round 1

Reviewer 1 Report

The study analyzed the association between a dietary pattern rich in high-potassium foods and serum potassium levels in hemodialysis patients. The study was well-carried and current form; I have some issues that need to be addressed.

The title of the manuscript should be reformulated. The findings presented in the manuscript did not support the potassium intake as a non-predictor of high serum potassium. The authors evaluated an association between potassium intake and serum. Predictive analyses were not carried out in the study. In my opinion, the authors assessed the association between adherence to the DASH diet and serum potassium concentration.

The introduction is clear and well-carried.

As age, gender, and BMI differs among groups (T1, T2, and T3), The authors should use an ANCOVA test instead of a one-way ANOVA.

The authors should add the medications used by the patients in the results section. Many of these drugs are potassium depletors and could be a study bias.

The potassium concentration in replacement solutions was similar among groups? This could be a bias of the study.

Is there a possibility of assessing urinary potassium in this cross-sectional study?

In the discussion, the authors highlight a threshold of 3000 mg/day of potassium as a reference to renal nutrition. Then, the authors report a mean potassium intake of 2765 mg/day and affirm that higher dietary potassium intake was not associated with more elevated serum potassium levels. What would be a high potassium intake for the authors? This isn’t very clear.

The relationship between dietary potassium intake and potassium levels could be improved in the discussion section. Please, see DOI: 10.3390/nu13082678, DOI: 10.1093/ndt/gfaa194.

Reviewer 2 Report

The aim of this study was to analyze the association between a dietary pattern rich in high-potassium foods (DASH) and serum potassium levels in HD patients. They found that a DASH dietary pattern, which is rich in potassium, is not associated with increased serum potassium levels in HD patients.

I would like to highlight that this is a very important paper in the field of chronic kidney disease management. I have a couple of minor comments:

1/ Did patients received dietary counseling prior to the study? Did those patients were already on a (low potassium) diet? If so, did all patients received a standardized dietary assessment/follow-up? In my opinion, the reader should be aware. As the majority of the patients had a potassium intake <3000, it seems that most of them were following a diet. 

2/ Which sources were the major contributions of total potassium intake in this study? This information might be of interest for the reader.

3/ the total potassium intake were for the majority of the patients <3000. It would be of interest to see if the absence of relationship between potassium intake of <3000 and >3000 is present in both groups. 

Round 2

Reviewer 1 Report

The authors addressed my concerns.